# Addressing Challenges in Diagnosis, Differential Diagnosis, and Treatment of Pemphigus: A Case Series

**DOI:** 10.3390/diagnostics13243633

**Published:** 2023-12-08

**Authors:** Zulfa Fidi Pranadwista, Etis Duhita Rahayuningtyas, Irna Sufiawati

**Affiliations:** 1Oral Medicine Residency Program, Faculty of Dentistry, Padjadjaran University, Bandung 40132, Indonesia; zulfa21004@mail.unpad.ac.id (Z.F.P.); etisduhita@gmail.com (E.D.R.); 2Department of Oral Medicine, Faculty of Dentistry, Padjadjaran University, Bandung 40132, Indonesia

**Keywords:** pemphigus foliaceous, pemphigus vulgaris, secondary syphilis-like pemphigus

## Abstract

Pemphigus is a rare autoimmune disease characterized by skin blisters and erosions, with or without mucosal involvement. The clinical presentation of pemphigus can resemble other bullous diseases, leading to challenges in diagnosis. This report aims to address the challenges in diagnosing and treating oral pemphigus. Three patients, ranging in age from 26 to 55 years, complained of a sore throat and mouth canker sores. Extra-oral examination revealed dry lips in case 1, while serosanguinolenta crust on the lip that bled easily was found in case 2. Intra-oral examinations in all cases showed multiple painful, sloughing-covered, erosive lesions on the entire oral mucosa. The histopathological examination of case 1 revealed pemphigus foliaceous, whereas cases 2 and 3 showed pemphigus vulgaris. Secondary syphilis-like pemphigus was given as a differential diagnosis in case 2 due to the histopathological changes not being specific. The patients were instructed to maintain oral hygiene and treated with corticosteroid, analgesic, antifungal, and anti-inflammation mouthwash, as well as vitamins and minerals. All cases showed improvement in oral lesions within 14 days to a month. In conclusion, pemphigus may mimic other bullous diseases, making diagnosis challenging. A comprehensive clinical and laboratory assessment is necessary to provide accurate diagnosis and treatment.

**Figure 1 diagnostics-13-03633-f001:**
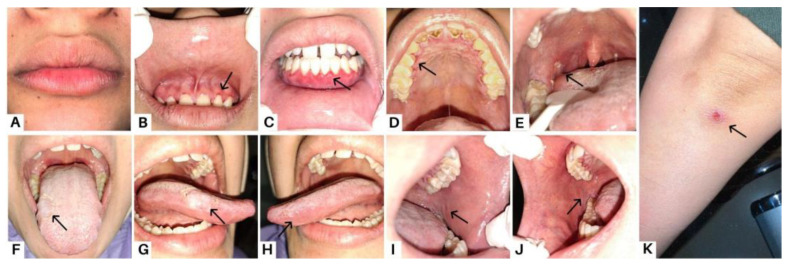
Case 1, a 23-year-old woman, complained of a sore feeling in her mouth starting one week prior, accompanied by a sore throat and painful swallowing for the past month. Clinical features of case 1 in the first visit are marked with black arrow as the dry lips (**A**); erosions in margin gingiva (**B**–**D**); an ulcer lesion on the right tonsillar pillar (**E**); sloughing-covered erosive lesions in the tongue and buccal mucosa (**F**–**J**); and bullous lesions that ruptured and left an ulcer area on the arms (**K**). A biopsy was performed three years prior on the skin lesion to examine the histological findings and obtain features of stratified squamous epithelium with the intact basement membrane, as well as superficial blister formation with detachment of the horny layer and some of the granular layer, which led to the diagnosis of PF. Pemphigus is a dermatosis that causes intraepidermal acantholysis, or keratinocyte separation [1]. Pemphigus is a series of blister diseases that are divided into two major subtypes: pemphigus foliaceous (PF) and pemphigus vulgaris (PV). Mucosal infiltration and suprabasal acantholysis distinguish PV from PF [2]. Blister disease is a condition where the body attacks healthy tissue by attaching autoantibodies to structural proteins in the mucous membranes and skin, which are components of the desmosome (desmocollins, desmogleins, and plakins) [3]. PF represents a superficial variant of pemphigus caused by antibodies against desmoglein 1 (Dsg 1), while Dsg 1 and Dsg 3 have been identified as the antigens responsible for PV. The patient was diagnosed with PV three months prior, after complaining of canker sores throughout the oral cavity followed by bullous lesions on the skin. Although the clinical profiles of PV and PF patients typically remain stable over time, transitions between the two have been recorded. Epitope spreading (ES), a fundamental inflammatory process that damages tissue by revealing latent immunological antigens and triggers a secondary autoimmune reaction, is one major explanation put forth for the transition from PV to PF or PF to PV, even though the mechanism underlying these changes is still not fully understood [2].

**Figure 2 diagnostics-13-03633-f002:**
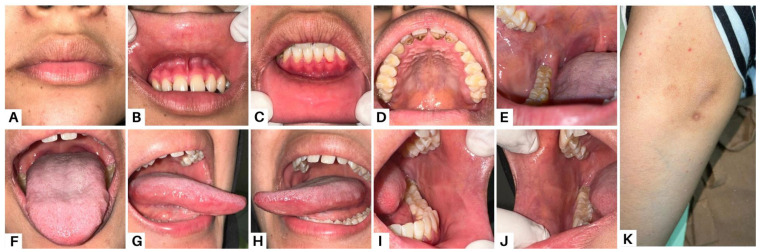
Methylprednisolone was given at a dose of 48 mg per day on the first visit. The patient was frequently checked every week and followed the instructions for using the drug given. Methylprednisolone therapy was continued with tapering of the dose every week. Furthermore, the patient was prescribed analgesic mouthwash, chlorhexidine gluconate 0.2% mouthwash, nystatin oral suspension, and petroleum jelly. Corticosteroids work quickly in PV with improvement seen within a few days, and new lesions stop appearing after two to three weeks. Re-epithelialization may take two months or longer. The dose of corticosteroids should be gradually decreased once the situation is under control, which is defined as the absence of new lesions with complete re-epithelialization of existing lesions. Although there are no fixed standards, the withdrawal process could take years because the reduction should be faster at the beginning and slower upon completion [4]. The risk of invasive fungal infection (IFI) increases when corticosteroids are given to immunocompromised patients. Antifungal medications may be used in conjunction with approaches that enhance host immune function or target interactions between the host immune system and the fungus [5]. An anti-HSV-1 IgG serological test was performed and revealed a positive result. We did not administer acyclovir therapy in case 1 and simply prescribed zinc and folic acid supplements since the ulcer lesions in the patient’s oral cavity had disappeared by the time the anti-HSV-1 IgG examination was revealed. Various minerals, including zinc and vitamin B forms in folic acid, play important roles in wound healing due to their biological effects [6]. After therapy was given, all lesions were improved in the fifth week (**A**,**B**,**D**–**K**), while a few erosions in the anterior gingiva still remain (**C**).

**Figure 3 diagnostics-13-03633-f003:**
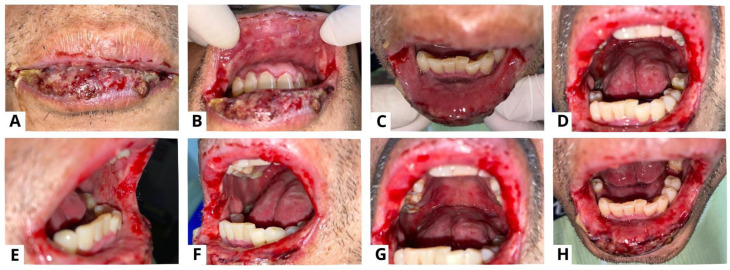
Case 2, a 55-year-old man, complained of canker sores throughout his mouth starting four months prior. Symptoms were felt to be becoming worse and spreading to the lips, to the point where he was having difficulty eating and opening his mouth. There were no similar lesions on other parts of the body, and the patient had never had this complaint before. The figure shows the clinical presentation of case 2 on the first visit. Extra-oral examination revealed sanguinolenta crust lesions on the lips and commissure of the lips that bled easily (**A**); intra-oral examination showed multiple sloughed-over erosive lesions on almost the entire surface of the oral mucosa (**B**–**H**). PV is a rare but potentially life-threatening autoimmune disease affecting the mucosa and skin. PV frequently involves the mucocutaneous sites, resulting in superficial blistering and persistent ulcerative lesions [7,8,9]. Syphilitic pemphigus is an uncommon syphilitic variation that typically manifests as bullous lesions on the palms and soles [10,11]. Because of this variety of clinical symptoms, syphilis is recognized as a great imitator [10,12]. PV is distinguished by painful erosions and the blisters are rarely whole, presumably due to their fragility and brittleness. Buccal and palatine mucosa, lips, and gingiva are the most impacted sites. The erosions are multiple with various sizes and irregular forms, and they peripherally extend and frequently show delayed re-epithelialization. Gingival involvement largely presents as desquamative gingivitis [4]. Pemphigus pathogenesis involves an autoimmune attack against the desmosomes and hemidesmosomes responsible for epithelial cell adherence. Desmogleins are a type of desmosomal cadherin that are reactive to autoimmune antibodies in PV [9,13,14]. The binding of antibodies leads to acantholysis and the formation of blisters which tend to rupture easily, resulting in painful erosions that impair one’s quality of life and cause secondary infections with increased morbidity and mortality [14].

**Figure 4 diagnostics-13-03633-f004:**
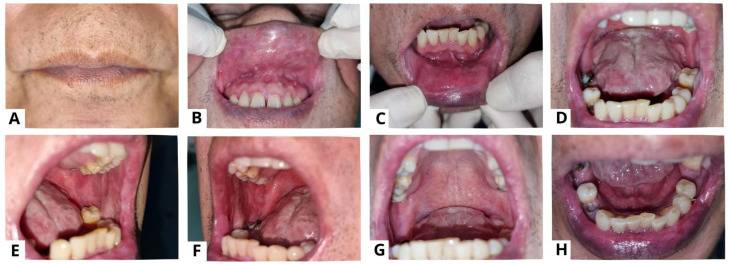
Clinical presentation of case 2 on the thirteenth week. All lesions were improved (**A**–**H**). Methylprednisolone therapy was given at a dose of 60 mg per day. The patient was also instructed to compress his lips using moistened gauze with 0.9% NaCL three times a day; use analgesic mouthwash; use hyaluronic acid 0.025% mouthwash; and use a formulated ointment consisting of a mixture of 0.5 mg dexamethasone, 2.5 mg lanoline, and 25mg petroleum jelly applied three times a day to the lips. The medication was used frequently and the patient reported that his complaints of oral cavity pain and bleeding on the lips decreased. The general condition of the patient also seemed to improve, and he gained weight. Nystatin oral suspension therapy was added from the fourth week to the seventh week, using 2 mL four times a day. Amoxicillin 500 mg was given three times a day for three days, and azathioprine (AZA) 50 mg was added twice a day. Methylprednisolone was maintained with tapering of the dose every week until it reached a daily dose of 12 mg on the twelfth visit. To minimize the negative implications, corticosteroids are usually tapered down before starting immunosuppressive medications [15]. One of the main adjuvants used in PV treatment is AZA. According to the recommendations of the European Dermatology Forum (EDF), it has been defined as a first-line adjuvant immunosuppressant [16].

**Figure 5 diagnostics-13-03633-f005:**
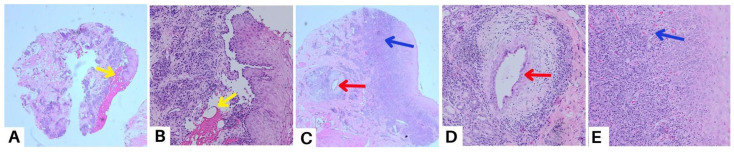
A punch biopsy of the buccal mucosal lesion was performed in the ninth week to obtain histopathological features and exclude other possible differential diagnoses. The results of the histopathological examination showed flattened epithelial cells with normal cell nuclei, some of which formed supra-basal blisters (yellow arrow) (**A**,**B**). The sub-epithelium showed endothelial “plump” cell appearance (red arrow) (**C**,**D**), and consisted of a fibro-collagenous connective tissue stroma covered with massive plasma inflammatory cells, lymphocytes, several histiocyte cells, and bleeding blood vessels (blue arrow) (**C**,**E**). These features point to a diagnosis of secondary syphilis like pemphigus, but as these histopathologic changes are nonspecific, we established a diagnosis of oral pemphigus vulgaris with a differential diagnosis of secondary syphilis like pemphigus. These histopathologic changes may occur after secondary infection in the form of blisters and ulceration. The diagnosis of pemphigus or other diseases that mimic pemphigus can be challenging for many reasons. In clinical practice, unusual or atypical manifestations of the bullous disease can be a diagnostic challenge, sometimes causing a delay in diagnosis, and so a variety of differential diagnoses depending on the clinical picture should always be considered [17]. Histopathological examination can help to diagnose pemphigus and distinguish it from other sub-epidermal bullous lesions, as acantholysis keratinocytes can be seen in several vesiculobullous diseases [4]. A biopsy was carried out in the ninth week due to a slow-healing lesion, which urged us to perform a punch biopsy on new lesions that reappeared on the buccal mucosa. Syphilis is difficult to diagnose due to the variety of dermatologic manifestations that might mimic other infections. Positive serologic test findings are used to diagnose secondary syphilis [10]. Secondary syphilis lesions might vary as much as their histopathologic features, like endothelial cell swelling, perivascular infiltrates with a preponderance of plasma cells, and epidermal psoriasiform hyperplasia, even though the alterations are frequently ambiguous [18]. Clinicians must continue to have a high index of suspicion for the diagnosis of syphilis in unusual dermatological presentations that are slow to respond to standard treatment. The patient in case 2 presented here was referred for serologic testing, and the Venereal Disease Research Laboratory (VDRL) revealed a positive result for syphilis. The VDRL or Rapid Plasma Reagin (RPR) test is routinely used in many laboratories as a screening test for syphilis because of its ease of performance, sensitivity, and low cost. Their sensitivity in the primary, secondary, and tertiary stages of syphilis is 60–70%, 100%, and 60–70%, respectively [19].

**Figure 6 diagnostics-13-03633-f006:**
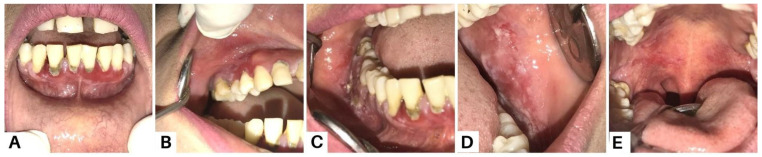
Case 3, a 35-year-old woman, reported experiencing canker sores for the past three months. Initially, she sought treatment from a general dentist, receiving antibiotics, analgesics, and anti-inflammatory mouthwash. However, these treatments did not lead to recovery. Intra-oral examination revealed multiple erosive lesions with sloughing, and desquamative lesions on the upper and lower gingiva (**A**–**C**), buccal mucosa (**D**), and palate (**E**). Oral lesions are the first symptom in 50–70% of PV cases and affect 90% of patients, often preceding widespread mucocutaneous involvement by several months [8]. The most commonly affected areas include the buccal and palatine mucosa, lips, and gingiva. In PV, the erosions are characterized by their multiplicity, varying sizes, irregular shapes, peripheral extension, and frequent delay in re-epithelialization. Gingival involvement primarily presents as desquamative gingivitis [4].

**Figure 7 diagnostics-13-03633-f007:**
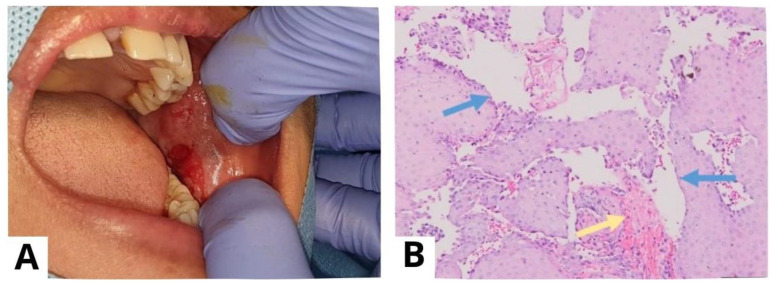
A punch biopsy was performed on the lesions on the left buccal mucosa (**A**). Prior to the procedure, the patient provided informed consent for a punch biopsy conducted under local anesthesia to establish a diagnosis. The histological examination (at 100× magnification) showed stratified squamous epithelium with suprabasal acantholysis (blue arrow) and fibrous connective tissue characterized by fibrocollagenous features with dense lymphocyte infiltration (yellow arrow), confirming the diagnosis of PV (**B**). For biopsy purposes, it is recommended to choose a recent blister (appearing less than 24 h ago) that can fit within a 4 mm punch or undergo a small fusiform excision, as PV blisters tend to rupture easily [4]. The results of a histopathological examination may indicate acantholysis and mild inflammatory infiltration. Due to acantholysis in the suprabasal layer, a single layer of basal keratinocytes remains attached to the dermal–epidermal basement membrane, resembling a “row of tombstones” [3]. The results of a histopathological examination may indicate acantholysis and mild inflammatory infiltration.

**Figure 8 diagnostics-13-03633-f008:**
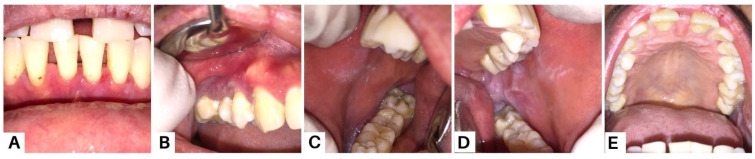
Clinical presentation of the oral mucosa in the eighth week; all lesions revealed improvements (**A**–**E**). Methylprednisolone was given at a daily dose of 16 mg, and was then gradually tapered over four weeks. The patient was also prescribed folic acid 1 mg, omeprazole 50 mg, vitamin D3 400 IU, and AZA 50 mg twice daily for the last two weeks. The patients were also instructed to maintain oral hygiene and use alcohol-free mouthwashes containing chlorhexidine gluconate 0.12%. Supra- and subgingival scaling were also carried out. Strategies for the management of PV aim to achieve and maintain complete remission with minimal side effects [20]. Systemic corticosteroid therapy is the most commonly used and well-established treatment due to its high efficacy and rapid control. We also administered AZA to the patient due to its efficacy as a corticosteroid-sparing agent in autoimmune bullous diseases, particularly in PV [4,15].

## Data Availability

The data presented in this study are available on request from the corresponding author. The data are not publicly available due to ethical restriction.

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
