# Peer review of "Addressing Challenges in Diagnosis, Differential Diagnosis, and Treatment of Pemphigus: A Case Series"

_diagnostics, 2023, doi:10.3390/diagnostics13243633_

Round 1
Reviewer 1 Report
There are multiple data that are not presented in the document and are relevant for the diagnosis of the conditions presented.
For example: In case 2, a diagnosis of secondary syphilis is made with VDRL, but in the event of a positive VDRL result (the authors should indicate the specific value), it must be confirmed with a specific treponemal test (for example: FTA- ABS). What treatment did patient 2 receive in relation to secondary syphilis?
Currently, the diagnosis of pemphigus requires knowing the serological values of anti-desmoglein 1 and 3 antibodies in peripheral blood (indirect immunofluorescen or immunoprecipitation). These data are basic and are not shown in the case descriptions.
These aspects should be resolved so that the cases can be accepted for publication.
Author Response
Dear Reviewer,
We greatly appreciate your valuable comments and suggestions. Please see the attachment.
Sincerely,
Irna Sufiawati
|
Response to Reviewer 1 Comments |
||
|
1. Summary |
|
|
|
Thank you very much for taking the time to review this manuscript. We have made some improvements according to the reviewer's suggestions and provided reasons for some of the feedback that we were unable to fulfill. Please find detailed responses below and revisions or corrections in the resubmitted manuscript. |
||
|
2. Questions for General Evaluation |
Reviewer’s Evaluation |
Response and Revisions |
|
Does the introduction provide sufficient background and include all relevant references? |
Must be improved |
Yes, in this article type (interesting images) there is no specific format for the introduction. However, we give a background of the three cases we presented including the relevant references and the purpose of this article. |
|
Are all the cited references relevant to the research? |
Can be improved |
Yes, all the references we used are relevant to this article and were published within the last ten years. |
|
Is the research design appropriate? |
Must be improved |
This article is a serial case report, so it does not require a research design. |
|
Are the methods adequately described? |
Must be improved |
This article is a case series report, so it does not require a method. |
|
Are the results clearly presented? |
Must be improved |
This article is a serial case report, so the improvement of the patient’s condition from the given therapy is clearly presented as the results of this article. |
|
Are the conclusions supported by the results? |
Must be improved |
The conclusion of this article is supported based on subjective, objective, and aid examination; as well as the results of the case management given. |
|
3. Point-by-point response to Comments and Suggestions for Authors |
||
|
Reviewer 1 |
||
|
Comments: There are multiple data that are not presented in the document and are relevant for the diagnosis of the conditions presented. For example: In case 2, a diagnosis of secondary syphilis is made with VDRL, but in the event of a positive VDRL result (the authors should indicate the specific value), it must be confirmed with a specific treponemal test (for example FTA- ABS). What treatment did patient 2 receive in relation to secondary syphilis? Currently, the diagnosis of pemphigus requires knowing the serological values of anti-desmoglein 1 and 3 antibodies in peripheral blood (indirect immunofluorescence or immunoprecipitation). These data are basic and are not shown in the case descriptions. |
||
|
Response: The patient in the second case had no prior suspicion of having syphilis because no risk factors were identified based on anamnesis, and the diagnosis of secondary syphilis was made as a consequence of the HPA examination results. Our hospital conducted the VDRL test, which resulted in a reactive result. Treponema pallidum hemagglutination assay (TPHA) examination and syphilis therapy were performed at another hospital in the patient's home city.
|
||
---------------------------------------------------------------------------
Reviewer 2 Report
This paper presents three cases of oral pemphigus, a relatively rare autoimmune and bullous condition in the oral cavity. A combination of clinical manifestations, histopathological findings, and immunological findings is necessary for the diagnosis of this disease. Unfortunately, evidences for the definitive diagnosis are poorly provided. The authors pointed out that this case resembled secondary syphilis only because of chronic inflammatory conditions in the histopathological picture, which is insufficient for the similarity to secondary syphilis. Therefore, some revisions are required for acceptance.
Queries and recommendations
1. It is necessary to remove “secondary syphilis mimicking pemphigus” from the title. The reason is explained below.
2. The authors emphasize that syphilis is a great imitator of pemphigus. However, bullous secondary syphilis is extremely rare, and it is not localized in the oral mucosa. Therefore, other bullous diseases, such as lichen plants pemphigoid, bullous pemphigoid, and others, should be discussed as differential diagnoses in this instance.
3. The results of anti-desmoglein indirect immunofluorescence, enzyme-linked immunosorbent assay, and direct immunofluorescence staining should be depicted in all three cases.
4. Add a cutaneous manifestation to Figure 1 as in Figure 6.
5. Instead of Intra-oral pictures, histopathological pictures of case 1 should be displayed in Figure 1.
6. The histopathological feature of case 2 with intraepithelial blistering or supra-basal clefts is diagnostic for pemphigus vulgaris. Authors suspected secondary syphilis based on chronic inflammatory change with swollen endothelial cells (I cannot confirm this feature in Figure 4B), but these chronic inflammatory conditions can be brought by secondary infection. Secondary syphilis exhibits considerable histopathological variability and shows psoriasiform hyperplasia with superficial neutrophils, epidermal apoptosis, and exocytosis of neutrophils. A superficial and deep chronic infiltrate with numerous plasma cells and often endothelial swelling can be seen but not specific. In addition, this case lacks extra-oral bullous signs, which could exclude secondary syphilis.
Author Response
Dear Reviewer,
We greatly appreciate your valuable comments and suggestions.
Please see the attachment.
Sincerely,
Irna Sufiawati
|
Response to Reviewer 2 Comments |
||
|
1. Summary |
|
|
|
Thank you very much for taking the time to review this manuscript. We have made some improvements according to the reviewer's suggestions and provided reasons for some of the feedback that we were unable to fulfill. Please find detailed responses below and revisions or corrections in the resubmitted manuscript. |
||
|
2. Questions for General Evaluation |
Reviewer’s Evaluation |
Response and Revisions |
|
Does the introduction provide sufficient background and include all relevant references? |
Must be improved |
Yes, in this article type (interesting images) there is no specific format for the introduction. However, we give a background of the three cases we presented including the relevant references and the purpose of this article. |
|
Are all the cited references relevant to the research? |
Can be improved |
Yes, all the references we used are relevant to this article and were published within the last ten years. |
|
Is the research design appropriate? |
Must be improved |
This article is a serial case report, so it does not require a research design. |
|
Are the methods adequately described? |
Must be improved |
This article is a case series report, so it does not require a method. |
|
Are the results clearly presented? |
Must be improved |
This article is a serial case report, so the improvement of the patient’s condition from the given therapy is clearly presented as the results of this article. |
|
Are the conclusions supported by the results? |
Must be improved |
The conclusion of this article is supported based on subjective, objective, and aid examination; as well as the results of the case management given. |
|
3. Point-by-point response to Comments and Suggestions for Authors
|
||
|
Reviewer 2 |
||
|
Comments 1: It is necessary to remove “secondary syphilis mimicking pemphigus” from the title. The reason is the authors emphasize that syphilis is a great imitator of pemphigus. However, bullous secondary syphilis is extremely rare, and it is not localized in the oral mucosa. Therefore, other bullous diseases, such as lichen plants pemphigoid, bullous pemphigoid, and others, should be discussed as differential diagnoses in this instance. |
||
|
Response 1: Agree. To highlight this point, the title of the article has been modified to "Addressing the Challenge in Diagnosis, Differential Diagnosis and Treatment of Pemphigus: A Case Series". |
||
|
Comments 2: The results of anti-desmoglein indirect immunofluorescence, enzyme-linked immunosorbent assay, and direct immunofluorescence staining should be depicted in all three cases. |
||
|
Response 2: Thank you for pointing this out. We agree with this comment. Therefore, the examinations mentioned were not carried out due to the patient's financial constraints and the test was not covered by government health insurance. |
||
|
Comments 3: Add a cutaneous manifestation to Figure 1 as in Figure 6. |
||
|
Response 3: Thank you for the suggestion, the picture of the cutaneous manifestation in Case 1 has been added in Figure 1 and Figure 2. |
||
|
Comments 4: Instead of Intra-oral pictures, histopathological pictures of case 1 should be displayed in Figure 1. |
||
|
Response 4: Thank you for pointing this out. We agree with this comment. Therefore, the patient in Case 1 had a biopsy performed in another hospital three years prior, thus we were just given the results of the examination report [explained in line 43] and not a histopathological picture. |
||
|
Comments 5: The histopathological feature of case 2 with intraepithelial blistering or supra-basal clefts is diagnostic for pemphigus vulgaris. Authors suspected secondary syphilis based on chronic inflammatory change with swollen endothelial cells (I cannot confirm this feature in Figure 4B), but these chronic inflammatory conditions can be brought by secondary infection. Secondary syphilis exhibits considerable histopathological variability and shows psoriasiform hyperplasia with superficial neutrophils, epidermal apoptosis, and exocytosis of neutrophils. A superficial and deep chronic infiltrate with numerous plasma cells and often endothelial swelling can be seen but not specific. In addition, this case lacks extra-oral bullous signs, which could exclude secondary syphilis. |
||
|
Response 5: Thank you for your advice. Furthermore, we concluded it as secondary syphilis based on the histopathology examination results, although secondary syphilis has a very varied clinical picture. The combination of interstitial inflammation, endothelial swelling, irregular acanthosis, and elongated rete ridges should raise the possibility of syphilis, even when no clinical suspicion exists. In addition, the condition of syphilis is also supported by the results of the VDRL screening that we conducted and obtained reactive results. |
||
|
4. Response to Comments on the Quality of English Language |
||
|
Point 1: The reviewer said that he/she was not qualified to assess the quality of English in this paper. |
||
|
Response 1: Thank you. |
||
|
5. Additional clarifications |
||
|
None. |
||
==================================================
Round 2
Reviewer 1 Report
The authors have made the proposed changes.
Author Response
Dear Reviewer,
On behalf of all the authors, I sincerely appreciate your time and efforts in evaluating our paper, as well as the invaluable comments you provided. It is through your valuable and insightful feedback that we have been able to make possible improvements in the current version.
We are now submitting the revised manuscript for further consideration and potential publication in the journal.
Sincerely,
Irna Sufiawati
---------------------------------------------------------------------------
Reviewer 2 Report
This paper presents three cases of oral pemphigus, a relatively rare autoimmune and bullous condition in the oral cavity. Although this manuscript has been partially improved, it still has a severe inadequacy for acceptance. I can understand that the immunological examinations were not carried out due to the constraints of the patient's finances and your country's insurance system. The correct diagnosis of this disease requires histological examinations, particularly in case reports. To accept this case series, it is necessary to have histopathological pictures.
Queries and recommendations
1. In case 1, the authors are required to borrow HE sections from the doctor who performed the previous biopsy.
2. I am not able to agree that case 2 was similar to secondary syphilis due to histopathological changes because these histopathological changes are not specific to it. These changes occur after secondary infection of blistering and ulceration.
3. As I pointed out in the first review, readers cannot identify swollen endothelium and plasmacytic infiltration in the histopathological photographs of Fig. 5B and 5C, respectively. Authors should replace these with appropriate photographs.
4. You should add explanations of what the arrows attached to Fig. 5 represented.
5. Unless you provide histopathological pictures, Case 3 should be removed from this series.
Author Response
Dear Reviewer,
On behalf of all the authors, I am pleased to re-submit our revised manuscript entitled “Addressing the Challenge in Diagnosis and Treatment of Pemphigus and Secondary Syphilis Mimicking Pemphigus: A Case Series,” for publication in Diagnostics.
We sincerely appreciate the time and valuable feedback provided by the editors and reviewers for our initial revised manuscript. Based on their comments and suggestions, we have made the following updates:
- Regarding case 1, regrettably, we were unable to obtain histopathological anatomy (HPA) photographs of the lesions. The hospital where the patient was initially seen, which is not a Teaching Hospital, does not retain slides after examinations. Nevertheless, we have included the interpretation of the HPA analysis in the re-submission on the website.
- In response to your recommendation, we have removed case 3 and replaced it with another case. We have also included HPA images to accompany this case.
We have also prepared a point-by-point response, detailing how we have addressed the comments and suggestions made by the Reviewers in the response letter.
Thank you for your attention to our revisions. We are now submitting the revised manuscript for further consideration in the journal, and we eagerly await the outcome of your assessment."
Sincerely,
Irna Sufiawati